

# Adaptive tourism forecasting using hybrid artificial intelligence model: a case study of Xi'an international tourist arrivals

Shuxin Zheng[1] and Zhongguo Zhang[2]

[1] School of Economics and Business, Changzhou Vocational Institute of Textile and Garment, Changzhou, China
[2] College of Computer Science and Technology, Taizhou University, Taizhou, China

## ABSTRACT

Accurate forecasting of tourist demand is important to both business practitioners and government policymakers. In the past decade of rapid development of deep learning, many artificial intelligence methods or deep learning models have been built to improve prediction accuracy. But data-driven end-to-end deep network models usually require large data sets to support. For tourism forecasting, the sample is insufficient and many models are difficult to apply. In this article, we propose a novel hybrid model GM-LSTM, which combines the advantages of gray models and neural networks to achieve self-adaptive prediction with small samples. Specifically, the overall trend of tourism demand is captured by a first-order gray model and the non-linear residual fluctuation is characterized using a long short-term memory (LSTM) network with a rolling mechanism. The model is validated through a case study of up to 38 years of data on annual international tourist arrivals in Xi'an, China. The proposed GM-LSTM model achieved a predicted MAPE value of 11.88%, outperforming other time series models. The results indicate that our proposed hybrid model is accurate and efficient.

## INTRODUCTION

Tourism of the 21st century has become one of the fastest and largest growing economic sectors through continuous diversification and expansion in the world (*Makoni, Mazuruse & Nyagadza, 2023*). According to the change of tourism trend, both government bodies and industry practitioners need to adjust their plans considering staff, capacity and resources (*Tang, Xie & Hu, 2022*). Therefore, it is of great importance to develop an accurate and efficient method for tourism demand forecasting. In recent years many forecasting models have been developed to predict the tourist demand (*Andariesta & Wasesa, 2022*). These forecasting models can be classified into three categories: econometric models, time series models and artificial intelligence (AI) models. The econometric models are based on the economic variables such as tourist income level, tourism price and geographical factors. Although econometric models provide insights into the causal factors, it is sometimes difficult to collect detailed data of explanatory variables for tourism forecasting.

Corresponding author
Zhongguo Zhang,
zhangzhongguo@tzu.edu.cn

Alternatively, non-causal forecasting models including time series models and AI models have gained more popularity in the past decade. They have the ability to forecast the future tourism demand through learning the trends and patterns in history data.

The most widely used time series models are the different forms of autoregressive moving average (ARMA) models (*Athanasopoulos et al., 2011*). *Ma et al. (2016)* used an autoregressive integrated moving average (ARIMA) model to perform 36-month-ahead tourism forecasting and benchmarked eight different ARIMA models. The seasonal autoregressive integrated moving average model (SARIMA) was proposed by *Jungmittag (2016)* to solve the monthly air travel forecasting problem at Frankfurt Airport. *Chu (2008)* discussed the application of fractionally integrated ARMA model and forecasted Singapore's tourist arrivals. In general, these time series models show good forecasting accuracy for linear and seasonal modelling. However, they are less capable to characterize non-linear behaviors such as substantial market volatility (*e.g.*, 2008 global financial crisis) (*Sun et al., 2019*). Another difficulty is that the model specification and the data distribution are often required to follow some assumptions before implementation (*e.g.*, stationary process) (*Song et al., 2012*).

With the rapid development of fundamental theory and advanced hardware, AI-based models have provided more flexibility and capability in characterizing the non-linear patterns, with a potential to achieve better performance than standard time series models. They overcome the need of formal specification and are more compatible with various data distributions. Among the different AI models introduced to tourism forecasting research, artificial neural networks (ANNs) have drawn major attention by the ability to characterize non-linear complex systems. *Cho (2003)* used ANN to forecast the number of arrivals from different countries to Hong Kong. *Constantino, Fernandes & Teixeira (2016)* compared ANN methods with different input, activation methods and nodes in terms of forecasting tourism demand in Mozambique. A model based on long short-term memory (LSTM) neural network for predicting tourist traffic was proposed by *Li et al. (2018)* and experimentally validated to be effective.

However, the training of deep neural network models, which are primarily data-driven, usually requires a large amount of data. This means that the predictive performance of the model is greatly reduced when the amount of data is too small or the data annotation is incomplete. In contrast, the grey model (GM) is an AI method proposed to overcome the limitation of insufficient data from the perspective of grey theory (*Ju-Long, 1982*). It offers a grey dynamics model-based approach to deal with the problems of uncertainty with few observations or poor information which is "partial known, partial unknown" (*Liu & Forrest, 2007*). Due to its simplicity and efficiency, the gray model has been successfully applied to various fields including tourism forecasting. In the existing literatures of AI-based tourism forecasting, most studies focus on improving the accuracy of a single model. Only a few studies have explored the combination of different AI models. *Sun et al. (2016)* developed a Markov-chain grey model with Cuckoo search optimization algorithm for forecasting of annual foreign tourist arrivals to China. *Hadavandi et al. (2011)* presented a hybrid artificial intelligence model to develop a Mamdani-type fuzzy rule-based system for tourist arrivals forecasting. Several recent studies have explored to combine the grey

model and other AI models in other forecasting tasks. *Badi & Elghoul (2023)* proposed a Grey-ARAS approach to investigate the role of social media platforms in spreading fake news during COVID-19 pandemic. *Tutak & Brodny (2023)* developed grey relational analysis and multidimensional scaling methods in assessing the level of innovation potential of European Union member states. *Tesić et al. (2023)* developed LMAW-grey MARCOS model for selection of a dump truck. However, it remains a challenge to fuse different AI models in an optimal way to fully utilize their respective advantages. The combination of different AI models has shown the potential of more accurate and efficient tourism forecasting. However, the simple combination of different models does not necessarily lead to enhanced forecasting performance. More research is needed on the systematic framework of model fusion, as well as the theoretical justification.

In this article, a new hybrid GM-LSTM model is proposed to solve the tourist demand forecasting problem with few and non-linear observations. Based on a small-sample dataset (*e.g.*, annual tourism demand), an adaptive grey model is constructed to capture the main tourism tendency while the LTSM neural network is trained to characterize the residual fluctuation. This hybrid model combines the advantage of LSTM neural network for non-linear modelling and the efficiency of grey model. The rest of the article is organized as follows. The 'Method' presents the GM-LSTM framework and introduces related models. In 'Experiments' and 'Results', we apply the GM-LSTM model to the forecasting of annual international tourist arrivals to Xi'an, China from 1980 to 2018. Finally, conclusions and future directions are described in 'Discussion'.

# METHOD

## Method framework

Annual tourism demand typically exhibits a general trend with fluctuations, which reflect observations of complex dynamic systems. Factors like tourism price, exchange rate, and security can influence annual tourism demand. A visual representation of this framework is shown in Fig. 1. The time sequence is divided into fixed-length segments using a rolling window. The tendency is then extracted from each segment by a first-order gray model. The residual fluctuations are decomposed in the time series. The LSTM mechanism is used to predict the residual fluctuation, which has been learned from historical data. The prediction is made one step ahead, and this process can be iterated for multi-step ahead forecasting using a rolling window. To enable adaptive prediction that captures real-time system characteristics, Fixed-length time series are selected by a rolling window algorithm, on the basis of which analysis and prediction tasks are implemented. Our hybrid framework involves several steps. Firstly, we extract the tendency $\hat{x}$ of the data within the rolling window using a first-order gray model. Further, the residual fluctuations $\hat{\varepsilon}$ are decomposed out in the time series. In addition to this, these fluctuations are fitted by an LSTM neural network. Having learned from the historical data, we can then make one-step-ahead predictions, which can be iterated for multi-step-ahead forecasting using the rolling window approach. The models and related algorithms involved in this framework are described in detail in the following subsections.

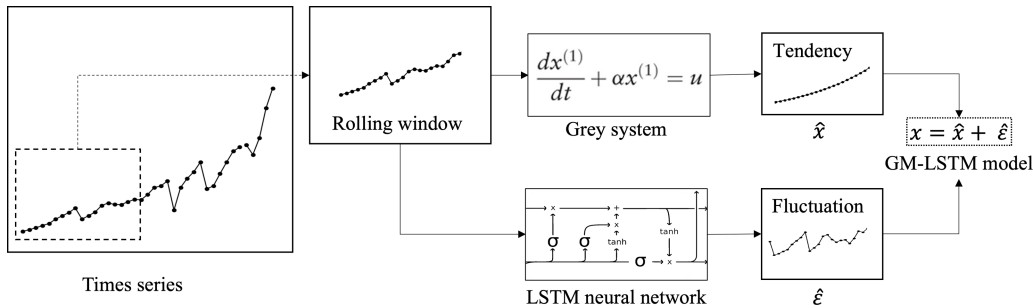

**Figure 1** **Overview framework of hybrid GM-LSTM model.** The whole time sequence is sliced into fixed-length segments using a rolling window. For each segment, the tendency is extracted using first-order gray model. Then the residual fluctuation is decomposed from the time series and is to be predicted using LSTM neural network. After learning from the history data, one-step-ahead prediction can be made and iterated for multistep-ahead forecasting using a rolling window.

## Gray model

Gray models are constructed for application to complex systems and short-term tourism forecasting through differential equation methods. In grey theory, $GM(n, m)$ model is defined, where n is the order of the difference equation and m is the number of variables. The $GM(1, 1)$ model, which utilizes first-order differential equations for univariate forecasting, is the most commonly used. Assume that the original time series is

$$x^{(0)} = [x^{(0)}(1), x^{(0)}(2), \ldots, x^{(0)}(j), \ldots, x^{(0)}(t)] \tag{1}$$

where $GM(1, 1)$ procedure forecasts the value $x^{(0)}(t+1)$ using the gray model, where $x^{(0)}(j)$ represents the $j$-th observation. And $t$ is the total number of training data. The steps are as follows:

Construction of a sequence of first-order cumulative generation operations (1-AGO):

$$x^{(1)} = [x^{(1)}(1), x^{(1)}(2), \ldots, x^{(1)}(j), \ldots, x^{(1)}(t)] \tag{2}$$

where $x^{(1)}(k) = \sum_{j=1}^{k} x^{(0)}(j)$, for $k = 1, 2, \ldots, t$. With this step we can obtain a smoother time series for the purpose of predicting the trend of the data.

Calculate the first-order differential equation:

$$\frac{dx^{(1)}}{dt} + ax^{(1)} = u \tag{3}$$

where $a$ represents the developing coefficient. $u$ is the gray input.

Use ordinary least squares (OLS) to estimate the parameters $a$ and $u$, with the solution being:

$$\begin{bmatrix} a \\ u \end{bmatrix} = (\mathbf{B}^T \mathbf{B})^{-1} \mathbf{Y}_N \tag{4}$$

where

$$B = \begin{bmatrix} -0.5(x^{(1)}(1)+x^{(1)}(2)) & 1 \\ -0.5(x^{(1)}(2)+x^{(1)}(3)) & 1 \\ \dots & \dots \\ -0.5(x^{(1)}(t-1)+x^{(1)}(t)) & 1 \end{bmatrix} \tag{5}$$

and

$$Y_N = [x^{(0)}(2), x^{(0)}(3), \dots, x^{(0)}(t)]^T. \tag{6}$$

Forecast model.

$$\hat{x}^{(1)}(t+1) = \left(x^{(0)}(1) - \frac{u}{a}\right)e^{-at} + \frac{u}{a} \tag{7}$$

and

$$\hat{x}^{(0)}(k) = \hat{x}^{(1)}(k+1) - \hat{x}^{(1)}(k) \tag{8}$$

where $\hat{x}^{(0)}(k)$ indicates the predicted value at time $k$.

## Rolling window scheme for gray model

Despite its ability to perform well with small samples, $GM(1,1)$ is limited in capturing real-time trends as it relies on the entire time-series for training. To overcome this issue and improve forecasting accuracy, a rolling window scheme (*Akay & Atak, 2007*) can be implemented. The approach involves training the model using only the most recent l observations, where l is the window size. This means that instead of using the entire time-series, $[x^{(0)}(t-l), x^{(0)}(t-l+1), \dots, x^{(0)}(t)]$ is the basic work of the prediction $\hat{x}^{(0)}(t+1)$. The optimal window size 1 is computed by a grid search algorithm. The algorithm measures and evaluates the prediction performance using different window sizes.

## Brief review on LSTM neural network

In 1997, Hochreiter and Schmidhuber proposed LSTM as a solution to the convergence issues that traditional neural networks face in time sequence prediction (*Hochreiter & Schmidhuber, 1997*). LSTM is a type of recurrent neural network (RNN) and its general structure is illustrated in Fig. 2A. RNNs excel in processing sequence data by incorporating the previous 'memory' of each state as input. The LSTM model comprises three parts: the input layer, the hidden layer, and the output layer. The inputs consist of $x$ data from the previous $T = [x(t-l), x(t-l+1), \dots, x(t)]$ historical moments. The output represents the target at the $(t + 1)$-th timestep, denoted as $y(t + 1)$. $H = (h_1, h_2, \dots, h_j, \dots, h_n)$ represents the neurons in hidden layer, where $h_j$ refers to the $j$-th neuron in the layer. The weight parameter carried by the hidden layer is represented by $W_{hh}$. $W_{ih}$ denotes the weight parameter between the input layer and the hidden layer. By the same token, the weight between the hidden layer and the input layer is represented by $W_{ho}$. The model is iteratively computed as follows:

$$h_n = \sigma(W_{ih}T + W_{hh}h_{n-1} + b_h) \tag{9}$$

$$y(t+1) = W_{ho}H + b_y \tag{10}$$

(A)

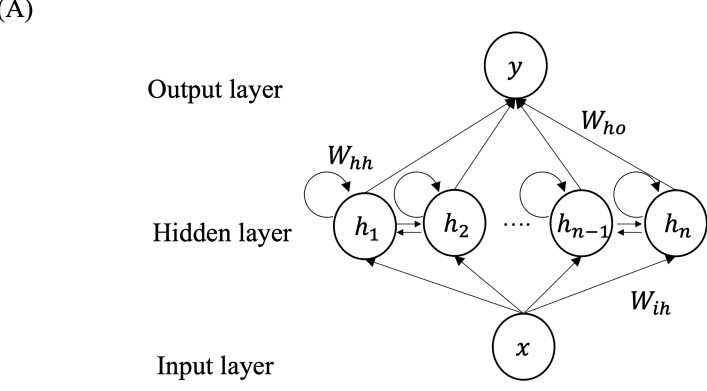

(B)

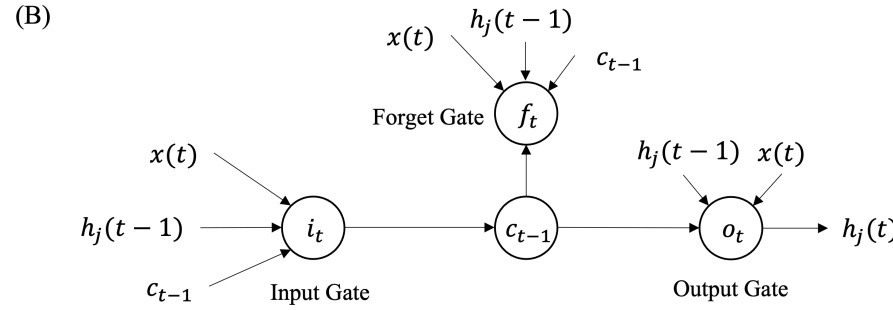

**Figure 2 Overview structure of the LSTM model (A) and structure of each LSTM neuron (B) for time series prediction.** LSTM is a type of RNNs consisting of three layers: input layer, hidden layer and output layer. Each neuron of the LSTM network includes input gate, output gate and forget gate to remember both long-range and short-range memory.

where $b_h$ refers to the size of the bias vector in the hidden layer. And the bias vector of the output layer is indicated as $b_y$.

Figure 2A illustrates that each neuron is comprised of three gates: the input gate, forget gate, and output gate, as depicted in Fig. 2B. The input gate selects the input information for the current cell state $c_t$ from $x(t)$, $c_{t-1}$, and $h_j(t-1)$. The forget gate decides which information should be discarded based on the current input $x(t)$, the previous cell state $c_{t-1}$, and the previous output of the j-th neuron $h_j(t-1)$. The value of the current cell output gate is determined by the input size of $c_t$, $h_j(t-1)$ and $x(t)$. The input gate $i_t$ and forget gate $f_t$ are computed as follows:

$$i_t = \sigma \left( W_{ix}x(t-1) + W_{ic}c_{t-1} + W_{ih}h_j(t-1) + b_i \right) \tag{11}$$

$$f_t = \sigma \left( W_{fx}x(t) + W_{fc}c_{t-1} + W_{fh}h_j(t-1) + b_f \right). \tag{12}$$

The calculation of the neuron's updated state is performed as follows:

$$c_t = f_t \cdot c_{t-1} + I \cdot g \left( W_{ci}x(t) + W_{ch}h_j(t-1) + W_{cc}c_{t-1} + b_c \right). \tag{13}$$

The computation of the output gate is:

$$o_t = \sigma \left( W_{ox}x(t) + W_{hh}h_j(t-1) + W_{oc}c_{t-1} + b_o \right) \tag{14}$$

where $W_{ab}$ represents the value of the weight matrix from layer $a$ to layer $b$. In addition to that, $\sigma(.)$ denotes the sigmoid activation function, and each bias term with subscript is the bias vector. The weights of LSTM model are updated using back-propagation algorithms with training samples of the time series. LSTM networks utilize a gating mechanism to implement a selective memory function, which enables them to better handle time sequence prediction tasks compared to conventional neural networks.

## GM-LTSM hybrid model

The primary objective is to capture the general trend using the gray model and represent the non-linear residual fluctuations with the LSTM model. For the training phase, we utilize the complete time series data $[x(1), x(2), \ldots, x(t)]$. In line with 'Gray Model', we start by extracting training samples $S_k$ using a rolling window of length l, for one-step prediction.

$$S_k = [x(k), x(k+1), \ldots, x(k+l)], k = 1, 2, \ldots, t-l-1. \tag{15}$$

where $S_k$ is the $k$-th training sample.

To forecast the next data point $x(k+l+1)$ for each sample, a hybrid model combining $GM(1,1)$ and LSTM is employed. The time series data within the rolling window is first fitted using the $GM(1,1)$ model described in 'Method Framework', which produces a prediction $\hat{x}(k+l+1)$. However, as the gray model fails to capture fluctuations, a LSTM model is used to refine the prediction by incorporating the fluctuation. To calculate the residual error of the one-step-ahead prediction by the gray model for the $k$-th rolling window, we employ the following formula:

$$\varepsilon(k+l+1) = x(k+l) - \hat{x}(k+l+1), k = 1, 2, \ldots, t-l-1. \tag{16}$$

During the prediction stage, we first collect the time series $S_{t-l} = [x(t-l), x(t-l+1), \ldots, x(t)]$ for the latest rolling window to predict the next data point $\hat{x}(t+1)$ using gray model. Additionally, we use LSTM to predict the corresponding residual fluctuation $\hat{\varepsilon}(t+1)$ based on the latest sample series. Thus, the predictions of these two machine learning models are properly combined and we obtain reasonable predictions.

$$\tilde{x}(t+1) = \hat{x}(t+1) + \hat{\varepsilon}(t+1). \tag{17}$$

The same process used for prediction can be iteratively applied to make predictions for subsequent time series at multiple time steps ahead while reducing their weight.

## ARIMA model

In order to evaluate the performance of our hybrid AI model, we have included this linear model for comparison purposes. When it comes to forecasting annual tourism demand, a non-seasonal ARIMA model with the form $ARIMA(p, d, q)$ is typically employed. Among them, $p$ represents the auto-regressive term, $d$ is the integrated term and $q$ is the moving-average term. The general formula can be expressed as follows:

$$\left(1 - \phi_1 B - \phi_2 B^2 \ldots - \phi_p B^p\right)(1 - B)^d x_t = \left(1 - \theta_1 B - \theta_2 B^2 - \ldots - \theta_q B^q\right)\varepsilon_t. \tag{18}$$

The data and random error terms at time $t$ are represented as $x_t$ and $\varepsilon_t$, respectively. The backward shift operator is denoted as B, and it is used to shift a time series backward by one

period. Specifically, $Bx_t = x_{t-1}$. The difference levels are denoted as d. The autoregressive coefficients are represented as $\phi_1, \phi_2, \ldots, \phi_p$, and they capture the effect of previous values of the time series on its current value. The moving-average coefficients are represented as $\theta_1, \theta_2, \ldots, \theta_q$, and they capture the effect of past forecast errors on the current value of the time series.

## EXPERIMENTS

### Dataset description

According to the latest data, Xi'an is one of the most famous historical and cultural cities in China and one of the world's most famous tourist destinations. Xi'an has a very large number of visitors every year, reaching over 360 million in 2019. For the dataset, we collected the number of international visitors to Xi'an from 1980 to 2018 as raw data. This information, which has been collected by the government of Xi'an and is presented in Table 1, will be used to assess the accuracy of our forecasting model. The data from 1980 to 2013 are used as the training set and the data from 2014 to 2018 are test set. Test data are {Year: Arrivals}: {2014: 124.23; 2015: 110.72; 2016:133.88; 2017:175.13; 2018:202.75}.

### Evaluation

In this study, the data spanning from 1980 to 2013 was utilized as the training set, while the most recent five years from 2014 to 2018 were reserved as the test set to assess the efficacy of various models.

To evaluate the models' performance, three commonly employed metrics were employed: mean square error (MSE), absolute mean error (MAE), and mean absolute percentage error (MAPE). These metrics are defined as follows:

$$MSE = \frac{1}{n} \sum_{i=1}^{n} (x_i - \hat{x}_i)^2 \tag{19}$$

$$MAE = \frac{1}{n} \sum_{i=1}^{n} |x_i - \hat{x}_i| \tag{20}$$

$$MAPE = \frac{1}{n} \sum_{i=1}^{n} \left| \frac{x_i - \hat{x}_i}{x_i} \right| \times 100\%. \tag{21}$$

### Model implementation

The models were coded and assessed in Python 3.7, which is open-source software. Here are the implementation details for each model:

*Gray Model.* First, the classical $GM(1, 1)$ gray model was used to adaptively train the entire training dataset and to make predictions on the test set. Then we base again on the rolling

**Table 1** International tourist arrivals in Xi'an from 1980–2018 (Unit: ten thousand). The data from 1980 to 2013 are used as the training set and the data from 2014 to 2018 are the test set.

| Year | Arrivals |
| --- | ---: |
| 1980 | 4 |
| 1981 | 6.71 |
| 1982 | 9.09 |
| 1983 | 12.38 |
| 1984 | 15.13 |
| 1985 | 21.15 |
| 1986 | 25.78 |
| 1987 | 30.15 |
| 1988 | 36.58 |
| 1989 | 21.2 |
| 1990 | 25.88 |
| 1991 | 31 |
| 1992 | 40.16 |
| 1993 | 43.5 |
| 1994 | 41.49 |
| 1995 | 41.35 |
| 1996 | 45.39 |
| 1997 | 48.53 |
| 1998 | 47.98 |
| 1999 | 55.41 |
| 2000 | 65.03 |
| 2001 | 67.2 |
| 2002 | 74.13 |
| 2003 | 33.66 |
| 2004 | 65.03 |
| 2005 | 77.56 |
| 2006 | 86.73 |
| 2007 | 100.01 |
| 2008 | 63.2 |
| 2009 | 67.29 |
| 2010 | 84.18 |
| 2011 | 100.23 |
| 2012 | 115.35 |
| 2013 | 121.11 |

gray model implemented on the training dataset using the rolling window method. The optimal window length was set to [4, 15], which was accurately determined in this step by performing a grid search based on the MAPE predicted in the previous step of the training set. Finally, the rolling window was slid to make predictions on the test set.

*GM-LSTM Model:* The GM-LSTM model is implemented through the original deep learning algorithm technology library in PyTorch 1.0, which operates through interface

calls. After training to obtain the gray model, we calculate the residual fluctuations of the training set. In addition to this, the same rolling window as the gray model is used to divide it into fixed length time series. This led to the generation of 22 samples for LSTM model training. One of the thankful things is that the LSTM model itself includes LSTM layers with 128 hidden neurons and an additional linear layer. This configuration is better suited for learning the mapping between historical time series and residual fluctuations. The Adam optimizer was employed with a learning rate of 0.0005 to update the weights of the LSTM network. And 200 epochs of training were conducted until the training error stabilized. Additionally, we also trained an LSTM model without GM components to directly predict tourism demand, following the same training process for LSTM networks, as a control model.

*ARIMA Model:* The ARIMA model can be implemented with the statsmodels library in Python. Checking the smoothness of time series data: ARIMA model requires the time series data to be smooth, *i.e.,* the mean and variance remain constant. If the data is not smooth, it needs to be differenced to make it a smooth series. The ARIMA model consists of three parameters: $(p, d, q)$. The combination of $(p, d, q)$ was determined by conducting a grid search within the range of 0 to 10 for each parameter. Determining the optimal values of these parameters can be identified using ACF (autocorrelation function) and PACF (partial autocorrelation function) plots. The parameter combination that resulted in the best Akaike information criterion (AIC) on the training set was selected, and the coefficients were fitted using the OLS algorithm. These selected parameters and the resulting coefficients from the fit will be used to predict the demand for tourism in the test set.

## RESULTS

After grid searching the AIC through the training set, we can determine the best combination of $(p, d, q)$ for the ARIMA model as (2,2,4). The model's diagnostic figures are depicted in Fig. 3, demonstrating a favorable fitting outcome. To achieve this result, the text has been condensed and rephrased for clarity. Figure 3 is the diagnostics plots of (2,2,4) performance on the training set. The standardized residual plot (top-left), the histogram (top-right) and the normal Q-Q plot (bottom-left) show that the residual error distributes around zero and follow a Gaussian distribution. The correlogram shows the values fall into the confidence interval. These observations suggest that the assumption of noise is appropriate.

The model performed well by accurately capturing both trend and fluctuations, with a low MAPE of 15.73%. However, it did not perform well on the test data, with a large forecasting error resulting in a high MAPE of 19.31%. The above results indicate that the ARIMA model is not suitable for generalization using only small samples. In contrast, the $GM(1, 1)$ model fitted on the entire training set effectively captures the overall trend needed for forecasting. Although no fluctuations are captured, this also leads to a higher MAPE on the training set with a value of 25.57%. Nonetheless, the $GM(1, 1)$ model demonstrated good generalization ability on the test set, with a lower forecasting error of 12.40%, as shown in Fig. 4. This suggests that the gray model may be a more suitable choice when

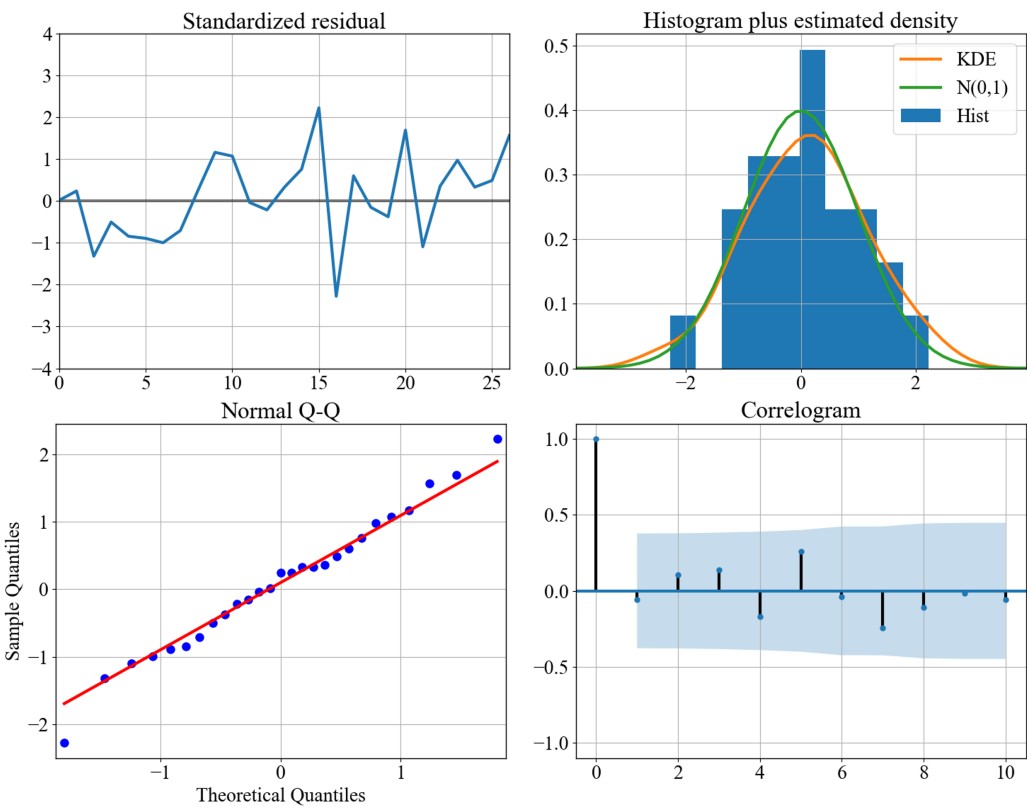

**Figure 3** **Diagnostics plots of performance on the training set.** The standardized residual plot (top–left), the histogram (top–right) and the normal Q-Q plot (bottom–left) show that the residual error distributes around zero and follow a Gaussian distribution. The correlogram shows the values fall into the confidence interval. These observations suggest that the assumption of noise is appropriate.

working with limited data. Models are trained using data from 1980 to 2013 and the performance is tested on the unseen data from 2014 to 2018. The original data is in plotted in black dot line. The results of ARIMA is plotted in blue dot line while the GM model is in red dot line. We conducted a comparison of the MAPE performance for one-step-ahead prediction on the training set using different rolling window lengths in the context of the rolling $GM(1,1)$ model.

Based on the results in Fig. 5, 12 was determined to be the optimal window length for this time series. Figure 6 is the comparison of one-step-ahead prediction of rolling $GM(1,1)$ and GM-LSTM hybrid model on training set and five-year-ahead prediction on the test set. A rolling window includes 12 successive time points. GM-LSTM model is trained using data from 1980 to 2013 and the performance of five-year-ahead prediction on test data from 2014 to 2018 is compared to GM model. The performance of one-step-ahead predication are also compared on the training set. Compared with the classical $GM(1,1)$ model, our proposed adaptive model exhibits superior performance. It obtains a predicted MAPE value of 11.88% and captures an enhanced growth trend. Nevertheless, our study

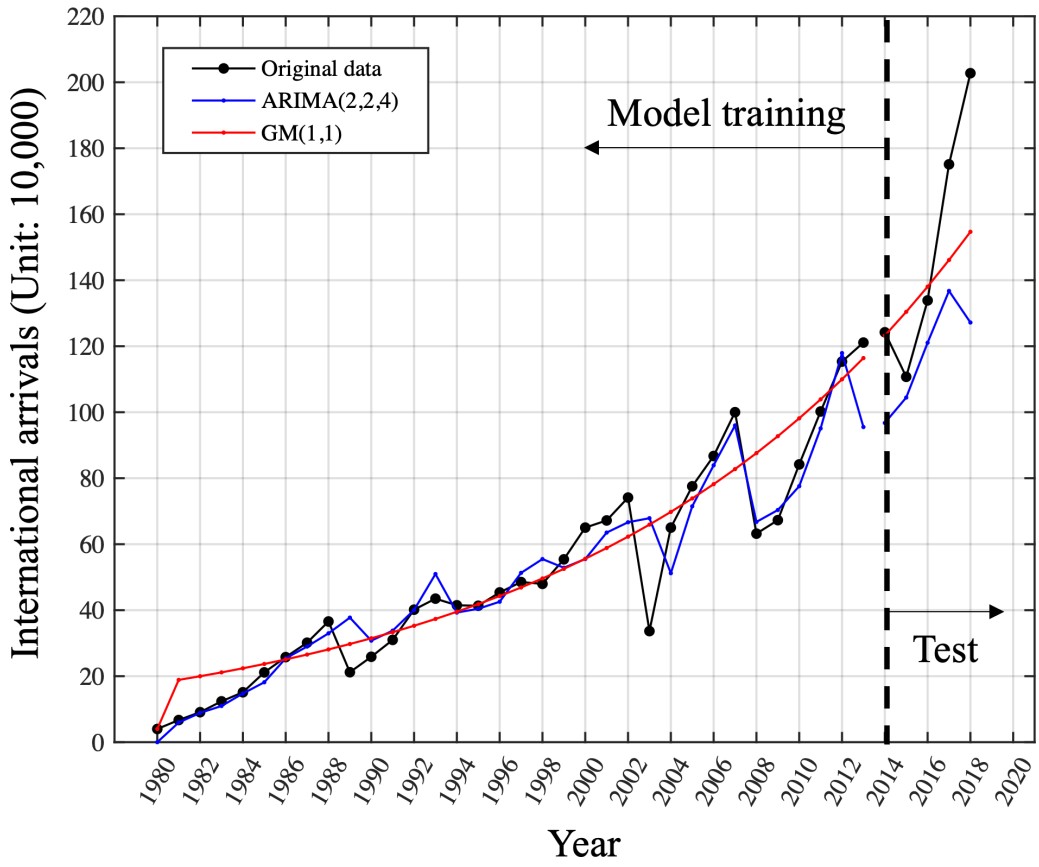

**Figure 4** **Comparison of ARIMA (2,2,4) and GM (1,1) on the training and test set.** Models are trained using data from 1980 to 2013 and the performance is tested on the unseen data from 2014 to 2018. The original data is plotted in the black dot line. The results of ARIMA is plotted in the blue dot line while the GM model is in the red dot line.

of the prediction results in Fig. 6 shows that the $GM(1,1)$ model still does not effectively characterize the fluctuations. Therefore, we need to innovate and improve the model.

## DISCUSSION

We can conclude from the experimental results that the LSTM mechanism in the GM-LSTM hybrid model can effectively represent the fluctuation trend of the training data. In addition, our proposed hybrid model approach outperforms the rolling $GM(1,1)$ model in both prediction of the training set and five-year prediction of the test set because it combines the advantages of the gray model and the neural network. This suggests that our hybrid AI model is both accurate and efficient. Interestingly, when we used a single LSTM model to predict tourism demand, we observed a significant drop in performance on the test set. Although the LSTM model was well-trained on the training set with a 13% MAPE for one-step-ahead prediction, it resulted in a much higher MAPE of 22% on the test set, indicating overfitting issues when the training data is limited. Table 2 presents a detailed performance comparison of the different models.

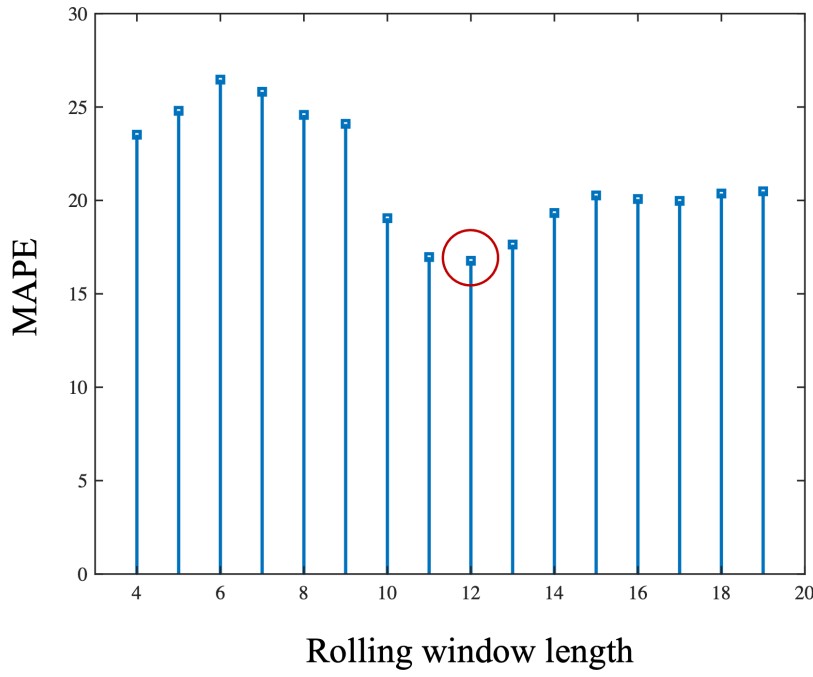

**Figure 5** **The length of rolling window and corresponding mean absolute percentage error (MAPE) of one-step-ahead prediction of rolling GM (1.1) on the training set.** The MAPE is lowest when choosing the rolling window length as 12.

**Table 2** **Forecasting performance of different models on the training and test set.** The methods with best evaluation metric are in bold.

| Models | MSE | MAE | MAPE | MSE | MAE | MAPE |
|---|---|---|---|---|---|---|
| | Training set (1980–2013) | | | Test set (2014–2018) | | |
| ARIMA (2,2,4) | **83.10** | **5.70** | **15.73** | 1267.95 | 32.10 | 19.31 |
| GM (1,1) | 115.55 | 7.80 | 25.57 | 711.69 | 20.39 | 12.40 |
| | One-step-ahead prediction | | | Test set (2014–2018) | | |
| Rolling GM | 219.36 | 10.20 | 16.77 | 453.34 | 18.08 | 11.88 |
| LSTM | **120.99** | **8.11** | 12.99 | 2786.00 | 35.88 | 22.33 |
| GM-LSTM | 123.23 | 6.69 | **12.43** | **290.65** | **14.17** | **10.24** |

The ARIMA and LSTM models are both data-driven and optimized based on the training data. However, in the case of forecasting tourism demand, the lack of training data limits the generalization ability of these models. On the other hand, the rolling GM model is model-based and training-free, as it is based on differential equations for dynamic evolution. This model performs well when the training sample is small, thanks to its assumptions and prior knowledge. Our hybrid method decomposes the non-linear signal into trend and fluctuation components, combining the strengths of both model-based and data-driven approaches. To our knowledge, this is the first hybrid intelligence method that addresses the small-sample problem in tourism demand forecasting. Furthermore, this framework can be extended with different model-based and data-driven techniques.

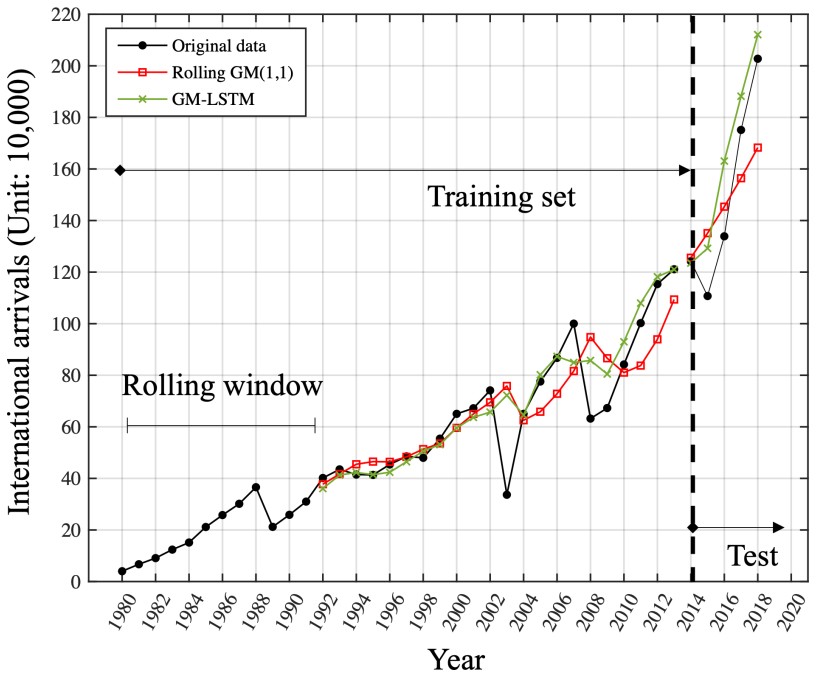

**Figure 6** **Comparison of one-step-ahead prediction of rolling GM (1.1) and GM-LSTM hybrid model on training set and five-year-ahead prediction on the test set.** A rolling window includes 12 successive time points. GM-LSTM model is trained using data from 1980 to 2013 and the performance of five-year-ahead prediction on test data from 2014 to 2018 is compared to GM model. The performance of one-step-ahead predication are also compared on the training set.

While the proposed GM-LSTM model demonstrates promising results, there remain several limitations and directions for future improvement. First, the current model only incorporates monthly tourist arrival data, while other potentially relevant factors like GDP, population, and seasonality are excluded. Expanding the model to multivariate inputs could enhance accuracy and provide more practical insights (*Doborjeh et al., 2022*). Secondly, the model hyperparameters and architecture are manually configured based on empirical analysis, which is suboptimal. Applying neural architecture search or Bayesian hyperparameter optimization may find better model configurations (*Kalliola, Kapočiūtė Dzikienė & Damaševičius, 2021*). Thirdly, the prediction interval of the hybrid model is not quantified, reducing its capability to deal with uncertainties. Developing the probabilistic version of GM-LSTM could generate predictive distributions rather than point estimates (*Liu et al., 2019*). Lastly, interpretability and transparency need to be considered when deploying black-box AI models. Techniques like attention mechanism (*Hsu, Liu & Tseng, 2019*) and model agnostic interpretation can improve model explanability (*Ismail et al., 2020*). The proposed GM-LSTM model provides a proof of concept for combining model-based and data-driven methods in tourism forecasting. Further research on multivariate modeling, automatic architecture design, uncertainty quantification, and interpretability will help transform this hybrid approach into a robust and practical forecasting tool.

# CONCLUSIONS

To address the challenge of predicting tourist demand in Xi'an, China with limited and complex data, this study introduces a novel approach—a hybrid GM-LSTM model. The proposed model demonstrates superior accuracy and real-time performance for forecasting tourist demand with sparse and nonlinear data. However, current research is still only based on univariate time series data predictions. Thus the models are not robust and generalizable enough and lack the ability to fully explain visitor behavior. Future studies should focus on developing multivariate forecasting models and conducting comprehensive analysis to gain a deeper understanding of tourist behavior prediction.

## Funding

This study is supported by the Research Foundation of Education Bureau of Jiangsu Province, China (Grant No. 19KJD120002). The funders had no role in study design, data collection and analysis, decision to publish, or preparation of the manuscript.

## Grant Disclosures

The following grant information was disclosed by the authors:
Research Foundation of Education Bureau of Jiangsu Province, China: 19KJD120002.

## Competing Interests

The authors declare there are no competing interests.

## Author Contributions

- Shuxin Zheng conceived and designed the experiments, performed the experiments, analyzed the data, authored or reviewed drafts of the article, and approved the final draft.
- Zhongguo Zhang performed the computation work, prepared figures and/or tables, authored or reviewed drafts of the article, and approved the final draft.

## Data Availability

    The model is available at Github and Zenodo:
    -https://github.com/siyukenny/Adaptive-tourism-forecasting-using-hybrid-artificial-intelligence-model.git.
    - Kenny. (2023). siyukenny/Adaptive-tourism-forecasting-using-hybrid-artificial-intelligence-model: Adaptive-tourism-forecasting-using-hybrid-artificial-intelligence-model (Version 111). Zenodo. https://doi.org/10.5281/zenodo.8252810.

## Supplemental Information

Supplemental information for this article can be found online at http://dx.doi.org/10.7717/peerj-cs.1573#supplemental-information.

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
