# Peer review of "Adaptive tourism forecasting using hybrid artificial intelligence model: a case study of Xi’an international tourist arrivals"

_PeerJ Computer Science, doi:10.7717/peerj-cs.1573_

## Round 0.1 · original submission · Major Revisions

Dear author,

Your paper has been reviewed by two reviewers who asked for major revisions of the paper. Please revise the paper according to comments by reviewers, mark all changes in new version of the paper and provide cover letter with replies to them point to point.

·

Basic reporting

Greetings,
The paper, which is called: Adaptive tourism forecasting using hybrid artificial intelligence model: a case study of Xi'an international tourist arrivals, is well written. English is at a good level. All the selections that a paper should have are in this paper. However, since the tables and figures were submitted as an annex to this paper, I cannot evaluate how they would look in the paper. Are they too big, do you need to change columns or rows and so on. The research objectives and problem are well defined. The results and discussion are well explained.

Experimental design

It is necessary to emphasize in the paper the aim and gaps that this research solves. The research problem and question are well defined. The research is well done. The methods used need to be better explained.

Validity of the findings

In this part, it is necessary to separate the results from the discussion. Make these special paper selections.

Additional comments

In order for the paper to be better, it is necessary to do the following:
First, far more references are needed to have this paper. This is especially the word in the Method selection since this paper does not have a literature review selection. I propose to extend Gray theory with the following papers:
- Using Grey-ARAS Approach to Investigate the Role of Social Media Platforms in Spreading Fake News During the COVID-19 Pandemic
- Progress towards the innovation potential of the European Union member states using gray relational analysis and multidimensional scaling methods
- Development of the MCDM fuzzy LMAW-grey MARCOS model for selection of a dump truck
And other papers. Do the same for each subtitle in this selection. The number of references must be at least over 30. So enrich this with references. When you separate the discussion, it is also necessary to provide references where this was also done and compare it with those studies. Then it is necessary to expand the conclusion with the limitations of this research and give directions for future research.

Reviewer 2 ·

Basic reporting

The authors proposed a novel hybrid model GM-LSTM, based on the advantages of gray models and neural networks to achieve self-adaptive prediction with small samples. To improve the paper quality, I propose the following improvements:
1. Please improve the abstract. Indicate what the results are.
2. I suggest starting a new paragraph from line 48. The sentence starts like this:” The most widely used time series models are the different forms….”
3. In line 66, it is unclear which author used ANN to forecast the number of arrivals from different countries to Hong Kong – Vincent or Li & Cao.
4. I propose to start a new paragraph from line 71.
5. Please, extend the literature background of the paper.

Experimental design

1. It would be nice to present the used sample in the paper.
2.The research motivation for proposing the given model should be better emphasized.

Validity of the findings

1. Please, explain the obtained results more in-depth. Also, compare the obtained results with similar research studies.

---

## Round 0.2 · accepted · Accept

Dear authors,

Both reviewers accepted your revised version of the paper.

·

Basic reporting

The paper is much better done. You should accept it now.

Experimental design

The paper is much better done. You should accept it now.

Validity of the findings

The paper is much better done. You should accept it now.

Additional comments

The paper is much better done. You should accept it now.

Reviewer 2 ·

Basic reporting

The authors have met all the requirements. Well-done.

Experimental design

The authors have met all the requirements.Well-done.

Validity of the findings

The authors have met all the requirements.Well-done.

Additional comments

The authors have met all the requirements.Well-done.